# Antimicrobial Susceptibility Profiles and Resistance Genes in Genus *Aeromonas* spp. Isolated from the Environment and Rainbow Trout of Two Fish Farms in France

**DOI:** 10.3390/microorganisms9061201

**Published:** 2021-06-01

**Authors:** Niki Hayatgheib, Ségolène Calvez, Catherine Fournel, Lionel Pineau, Hervé Pouliquen, Emmanuelle Moreau

**Affiliations:** INRAE, Oniris, BIOEPAR, 44300 Nantes, France; segolene.calvez@oniris-nantes.fr (S.C.); catherine.fournel@oniris-nantes.fr (C.F.); lionel.pineau@oniris-nantes.fr (L.P.); herve.pouliquen@oniris-nantes.fr (H.P.)

**Keywords:** *Aeromonas*, antibiotic-resistant bacteria, fish, environment, resistance genes

## Abstract

This study presents the occurrence and abundance of *Aeromonas* antibiotic-resistant bacteria (ARB) and genes (ARGs) isolated from water, biofilm and fish in two commercial trout farms before and one week after flumequine treatment. Wild (WT) and non-wild (NWT) strains were determined for quinolones (flumequine, oxolinic acid and enrofloxacin), oxytetracycline (OXY), florfenicol (FFN), trimethoprim-sulfamethoxazole (TMP) and colistin (COL), and pMAR (presumptive multi-resistant) strains were classified. Forty-four ARGs for the mentioned antibiotics, β-lactams and multi-resistance were quantified for 211 isolates. *Bla*SHV-01, *mex*F and *tet*E were the dominant ARGs. A greater occurrence and abundance of *tet*A2, *sul*3, *floR*1, *blaSHV*-01 and *mex*F were observed for NWT compared to WT. The occurrence of pMAR and NWT *Aeromonas* for quinolones, OXY, FFN, TMP, COL and ARGs depended on the *Aeromonas* origin, antibiotic use and the presence of upstream activities. Our results revealed the impact of a flumequine treatment on *Aeromonas* present on a fish farm through an increase in NWT and pMAR strains. The link between fish and their environment was shown by the detection of identical ARB and ARGs in the two types of samples. There appears to be a high risk of resistance genes developing and spreading in aquatic environments.

## 1. Introduction

A rise in antibiotic-resistant bacteria (ARB) and antibiotic-resistant genes (ARGs) has been reported in pathogenic, commensal and environmental bacteria over the last few years as a consequence of the wide use of antimicrobial agents to control human and animal infections [1,2,3]. The overuse of antimicrobial agents is a major source of antibiotic pollution in the environment [4,5,6]. Like other farmed species, aquatic animals may play a role in the selection and spread of resistant environmental and pathogenic bacteria [7,8,9,10,11]. In global aquaculture production, the most widely used antibiotics are from the trimethoprim/sulfonamide, quinolone and tetracycline families, which were found to be related to the development of ARB and ARGs, and more often, multidrug resistance strains [12,13]. Approximately 80% of antibiotics used in aquaculture, which are commonly applied as a feed supplement in pond water, enter the environment with their activity intact [14]. The overuse of antimicrobial agents is a major source of ARGs and antibiotic pollution in the environment. Some of the antibiotics administered in fish are excreted unchanged in feces and urine and discharged into rivers. This may lead to the contamination of surface water, and sometimes of water intended for human use, such as drinking water supplies [15,16,17].

The prolonged presence of antibiotics in raceway water, combined with high numbers of bacteria in the polybacterial matrices of biofilms and potential contamination of aquatic environments by pathogens of human and animal origin, could stimulate selective pressure on the exchange of genetic information between aquatic and terrestrial bacteria, and creates the potential risk of the development and spread of antibiotic-resistant bacteria and genes between fish, their environment and humans [18,19]. The passage of antimicrobial-resistant bacteria and resistance genes from fish and their environment to terrestrial livestock and humans could favor the survival and maintenance of ARB and the widespread emergence of drug-resistant pathogens in environmental reservoirs. Moreover, upstream aquaculture activities should be considered as a reservoir and the origin of ARB and ARGs in downstream animal and human facilities [6,16].

*Aeromonas* is a genus of Gram-negative bacteria belonging to the *Aeromonadaceae* family, and consists of a group of opportunistic environmental pathogens, with some species being able to cause disease in humans, fish and other aquatic animals [20,21]. They are autochthonous to aquatic environments and have been easily isolated from different kinds of water, such as rivers, lakes, ponds, estuaries, drinking water, groundwater, wastewater and sewage in various stages of treatment, and they may persist attached to biofilms on biotic or abiotic surfaces in environment ecosystems [22,23]. *Aeromonas* outbreaks are currently a common phenomenon in freshwater farmed fish. Some *Aeromonas* species, such as *Aeromonas salmonicida* sub *salmonicida*, are a pathogen agent of furunculosis, which is one of the most common diseases in salmonid farmed fish worldwide that causes important financial losses in the aquaculture industry [24,25]. In France, one of Europe’s biggest aquaculture producers of freshwater fish (39,500 tonnes in 2019), notably rainbow trout (*Oncorhynchus mykiss*), furunculosis has been fairly well controlled. However, recurring clinical cases were recently reported, particularly in the late spring and summer [26,27].

Previous studies have indicated the presence of *Aeromonas* in aquaculture systems with high levels of resistance to antibiotics and gene resistance determinants [7,8]. Some studies assessed the antimicrobial sensitivity of *Aeromonas* species that were isolated from farmed rainbow trout and their environment in which they were resistant to quinolones and fluoroquinolones, streptomycin, oxytetracycline, chloramphenicol, florfenicol, sulfamethoxazole-trimethoprim and β-lactams [28,29]. Multidrug-resistant *Aeromonas* gene-harboring strains like *sul*1, *tet*A and *floR* also have been detected in different species of farmed fish [11,30]. However, an analysis of the high diversity and abundance of *Aeromonas* ARGs and their antimicrobial sensitivity profiles due to antibiotic treatments has not yet been carried out in fish farms and their environment. Furthermore, studies on *Aeromonas*’s antimicrobial susceptibility remain rare, and no epidemiological cut-off values are currently available from the European Committee on Antimicrobial Susceptibility Testing (EUCAST) to interpret the minimum inhibitory concentrations (MICs) of *Aeromonas* spp. [30,31,32]. To understand the extent of ARG transmission in aquatic ecosystems, this study focused on the evolution of antimicrobial susceptibility (MIC) and resistance genes in environmental and fish *Aeromonas* isolated from two rainbow trout fish farms over a seven-month period that included episodes of furunculosis and an administration of antibiotics.

## 2. Material and Methods

### 2.1. Ethics Statement

This study was approved by the members of the Animal Experiment Ethics Committee of Oniris in France (CERVO-2020-6-V).

### 2.2. Description of the Farms

This study was carried out on two commercial rainbow trout fish farms (fish farms A and B) in France over seven months (February to August 2020). Both farms are fed by river water through an open water circuit system. The average water temperature in both fish farms was recorded at 10 ± 0.5 °C and 14 ± 0.5 °C in winter and summer seasons, respectively. These fish farms are composed of upstream ponds for juvenile trout and downstream ponds for larger trout until they reach the commercial weight. In this study, the large trout from 40 ± 5 cm/800 ± 200 g to 55 ± 5 cm/2000 ± 200 g were considered at the beginning and end of the study, respectively. The farms were chosen due to their recent history of furunculosis and antibiotic use. In August 2018, furunculosis was observed and treated on farms A and B using trimethoprim/sulfonamide and florfenicol antibiotics, respectively. Earlier, farm B had experienced furunculosis outbreaks that were treated with the same antibiotics and enrofloxacin (in July 2016 and 2017). Furthermore, farm B also administered the furunculosis autovaccine from the end of 2018 through February 2020, with the last dose given 10 days before the start of the study. No vaccination was carried out on farm A. These two fish farms were also selected based on their different environmental areas and biosecurity practices. Farm A, with 320 tonnes of production per year, was situated near other animal farms, including two other fish farms and several pig and cattle breeding sites located upstream of farm A. Farm B, with 110 tonnes of production per year, was situated in an isolated area without any other farms nearby.

### 2.3. Sampling

Fish farms A and B were monitored monthly over seven months to survey the health of the fish and the administration of antibiotics in the case of disease outbreaks. Monthly samples of fish, pond water and biofilm were taken from two existing downstream raceways close to the end section of the rearing ponds on each farm. These ponds were dedicated to this study and no fish were added to the pond water during the study period. Sampling on farm A was realised from February to August, while sampling on farm B was carried out from February to July (fish were commercially slaughtered in August). On farm A, two additional clinical samples were also taken following furunculosis episodes, one in May and another in July, which were collected one day before the start of the antibiotic treatment in July. One monthly sampling in August was then carried out one week after the end of the antibiotic treatment. The sampling schedule for April was cancelled in both farms due to health regulations related to the worldwide COVID pandemic.

All of the collected samples were transported for further bacteriological analysis under proper cold transport conditions on the day of the visit within 2–3 h of being taken.

### 2.4. Fish Samples

In total, 18 fish were sampled from each fish farm monthly and during the additional visits. The studied population was selected based on the maximum probability of isolating the *Aeromonas* bacteria from fish farms that were recently infected with furunculosis. The fish sample size for each pond (≤22,500 fish per raceway) was determined by considering that the expected frequency of furunculosis was 30% in the studied farms based on the analysis of Cannon and Roe [33,34]. Fish samples were autopsied and clinical lesions were recorded for each case. The samples of spleen, mucus from the posterior digestive tract, gills and skin mucus were dissected and cultured on Agar GSP (Merck KGaA, Darmstadt, Germany), the selective medium for detecting *Aeromonas* spp. [35].

### 2.5. Water Samples

One litre of pond water was collected from each study pond using a sterile water bottle. Each water sample was filtered in 10 parts (10 × 100 mL) using the filtration manifold system (Millipore, Darmstadt, Germany) through a cellulose nitrate membrane filter, 47 mm diameter, 0.22 μm pore size (Sartorius, Goettingen, Germany). The filter membrane then was placed in a Petri dish into which 1 mL of sterile normal saline solution was added. The bacteria were detached from the filter membrane via pipetting the sterile water on the membrane [36]. The solution then was diluted at 10^−1^ and 10^−2^. Then, 100 µL of each dilution was inoculated and thinly spread on Agar GSP (Merck KGaA, Darmstadt, Germany).

### 2.6. Biofilm Samples

Prior to the start of the study, biofilm experimental surfaces were created on a plastic structure and installed in each study pond. Each month, two biofilms were removed from each pond and taken for analysis. One biofilm surface was taken for the bacteriological analysis of the cumulative effects of previous months. Another biofilm surface was also collected from the previous month and then replaced with the biofilm surface of the following month. Each biofilm plate was placed in a sterile bottle filled with the corresponding pond water. The plastic biofilm surface (5 × 5 cm) was detached from the plate and put into the filtered sterile stomacher bag (177 × 302 mm) into which 10 mL of sterile normal saline solution was added. After being put in a mini-mixer (stomacher) (Lab-Blender 400, Leicestershire, UK), which operated at a speed of 230 rpm for 15 min, attached cells were removed from the biofilm surface into the stomacher bag. Using a sterile pipette, the filtered cells were aspirated from the stomacher bag and placed in a sterile tube [37]. The samples were diluted at 10^−2^ and 10^−3^, and then 100 µL of solution was inoculated and thinly spread on Agar GSP (Merck KGaA, Darmstadt, Germany).

### 2.7. Aeromonas spp. Isolation and Identification

All seeded samples isolated from fish, water and biofilm were incubated at 22 °C for 48 h. Then, up to five yellow colonies, which were often surrounded by a yellow zone (depigmentation of the GSP medium), were removed per fish organ, and two colonies from biofilm and water samples. Each isolated colony was subcultured in Agar GSP for 48 h at 22 °C in order to obtain the pure colonies [35]. After 48 h, the pure colony was inoculated in liquid medium (TSB) (Biokar, Beauvais, France) for 24 h at 22 °C. *Aeromonas* spp. were identified at the genus level using polymerase chain reaction (PCR) [38], and cultures were conserved in a cryopreservation tube at −80 °C. By considering the origin and morphology of colonies in order to avoid the cluster-forming units, up to three *Aeromonas* isolates per sample were selected for antimicrobial susceptibility testing. Then, up to two *Aeromonas* isolates per sample, depending on the isolate’s antimicrobial susceptibility profiles, were selected for antimicrobial resistance gene studies. These isolates were classified as healthy isolates when no episode of furunculosis or no antibiotic treatment were observed, furunculosis isolates when furunculosis occurred and treated isolates when followed by an antibiotic treatment.

### 2.8. Antimicrobial Susceptibility Test

The broth micro-dilution method (document M49-P) [39] was used to determine the MIC values of seven antimicrobial agents, namely flumequine, oxolinic acid, enrofloxacin, oxytetracycline, florfenicol, sulfamethoxazole-trimethoprim and colistin, for *Aeromonas* isolates. In this study, the antimicrobial agents were chosen based on their use in veterinary medicine, mainly in aquaculture, and human medicine against *Aeromonas* infections and the consideration of antimicrobial resistance profiles [5,40]. To prepare the antibiotic solutions, each antimicrobial agent at 20× concentration was first prepared with the recommended solvent. The antimicrobial solutions were then diluted 1:10 in adjusted BMH (Oxoid, UK). Afterwards, a series of doubling dilutions of each antimicrobial agent was prepared in BMH to obtain final concentrations of flumequine (0.016–256 µg/mL), oxolinic acid (0.016–64 µg/mL), enrofloxacin (0.016–64 µg/mL), oxytetracycline (0.016–512 µg/mL), florfenicol (0.25–32 µg/mL), trimethoprim-sulfamethoxazole (0.015/0.3–64/1216 µg/mL) and colistin (0.781–400 µg/mL) before the bacterial strains inoculations. Fifty microlitres of each solution were distributed in 96-well microplates (Corning^®^ 3367; 96 Wells, Costar, NY, USA). The positive and negative control wells received 50 µL of BMH used for the preparation of dilutions and then the microplates were stored at −20 °C. For all MIC assays, *Escherichia coli* ATCC 25,922 and *A. salmonicida* subsp *salmonicida* ATCC 33,658 were used as reference controls (document M49-P) [39].

Briefly, the overnight TSA (BIOKAR ref. BK047HA; Beauvais, France) cultures of *Aeromonas* spp. were incubated in BMH broth at 22 °C for 24 h. Then, *Aeromonas* spp. cultures were re-incubated for about 3–6 h in BMH broth at 22 °C with continuous agitation. These cultures were diluted in BMH at 1% to obtain a final concentration at approximately 10^6^ CFU/mL. The calibration of the inoculum was verified via bacterial enumeration. Fifty microlitres of inoculum suspension of each bacterial strain were mixed with 50 µL of each dilution of antimicrobial agents in a U-bottom assay microplate (Corning^®^ 3367-96 Wells, USA). The positive and negative control wells received 50 μL of BMH used for the inoculum suspension. Microplates were incubated under aerobic conditions at 22 °C for 24 h. MIC values were determined at the lowest concentration where no bacterial culture was observed after 24 h of incubation in accordance with the guidelines of the Clinical and Laboratory Standards Institute (CLSI) [39].

### 2.9. MIC and Presumptive Epidemiological Cut-Off Values (COWTs) Analysis

From the distribution of the MIC values, the minimum inhibitory concentration required to inhibit the growth of 50% (MIC50) or 90% (MIC90) of the strains, as well as presumptive epidemiological cut-off values (COWTs), were calculated.

COWTs were calculated using two methods [31], namely, the Kronvall and Turnidge methods [41,42]. For the Kronvall method, a fully automated and freely available Excel spreadsheet calculator (updated version, 2019) was used to apply the normalised resistance interpretation (NRI) (available at http://www.bioscand.se/nri, accessed on 18 January 2021). The Turnidge method was applied through an updated version (2020) of the ECOFFinder tool (available from the EUCAST website at https://www.eucast.org/mic_distributions_and_ecoffs, accessed on 18 January 2021, ECOFF95%, SOP10.1, accessed on 18 January 2021). In this study, the determination of COWTs (Kronvall and/or Turnidge) depended on the distribution of MIC values for each antibiotic for *Aeromonas* isolates.

Following the CLSI guidelines, microbial populations were separated into two interpretive categories: a wild-type population (WT), i.e., those with no mechanisms of acquired resistance or reduced susceptibility for the antimicrobial agent, and a non-wild-type population (NWT), i.e., those with presumed or known mechanisms of acquired resistance and reduced susceptibility for the antimicrobial agent. The number and percentage of the NWT were calculated by considering all Kronvall and/or Turnidge results. Multidrug resistance was defined as the absence of susceptibility to at least one agent in three or more antimicrobial categories [43]. In this study, the number of presumptive multi-antibiotic resistant *Aeromonas* (pMAR) was calculated for all environmental and clinical studied samples among five antimicrobial categories, including quinolone, tetracycline, sulfonamide, polymyxin and phenicol.

### 2.10. Detection and Relative Abundance of Aeromonas ARGs

DNA extraction was performed following the protocol of isolating Genomic DNA Gram-Negative Bacteria [44] using Wizard^®^ Genomic DNA Purification Kit (Promega, Madison, WI, USA) according to the manufacturer’s instructions, with added enzymatic and mechanical cell lysis steps. Afterwards, DNA was quantified using Thermo Scientific™ Spectrophotometers NanoDrop™ 2000/2000c (Fisher Scientific SAS, Illkirch-Graffenstaden, France) and then stored at −80 °C until use.

The presence of common ARGs was studied in relation to the antibiotic classes frequently used in both veterinary medicine, mainly in aquaculture, and human medicine against *Aeromonas* infections [5,40], including *qnr* and *aac(6′)-Ib* for fluoroquinolone; *dfr*A, *sul* and *str* for sulfonamide-trimethoprim; *mcr* for polymyxin; *tet*A, *tet*B, *tet*C, *tet*D, *tet*E, *tet*G and *tet*M for tetracycline; *floR* and *cat*A for phenicol; *bla-*CTX-M, *bla-*ACC, *bla-*DHA, *bla-*IMP, *bla-KPC*, *bla*SHV and *bla*CMY for β-lactams; *mex*F for multidrug ARGs.

In total, a set of 44 specific primer pair genes and three housekeeping genes, including *16S*-1, *16S*-2 rRNA and *rpo*B genes (Table 1), were selected to target sequence diversity within a gene [19,45,46,47,48]. A negative control (no DNA) was also considered in each quantitative PCR (qPCR) run. The qPCR amplification was performed via the “Human and Environmental Genomics” Platform (Rennes, France) using the Takara SmartChip Real-time PCR system (Takara, Mountain View, CA, USA), which runs a high-throughput, nanolitre-scale real-time PCR. The 5184-well plates with a reaction volume of 100 nL were filled with the SmartChip MultiSample NanoDispenser (Takara, Mountain View, CA, USA). The SmartChip MyDesign Kit (Takara, Mountain View, CA, USA) was used and the PCR cycling conditions were as follows: denaturation at 95 °C for 5 min, followed by 42 cycles that included denaturation at 95 °C for 10 s, annealing at 60 °C for 30 s and elongation at 72 °C for 30 s. A final round of denaturation–annealing was performed. The specificity of amplification was assessed through the analysis of the melting curve of each PCR product. The detection limit of amplification was set at a threshold cycle (C_T_) of 27 [49]. The relative abundance of each detected gene was calculated proportionally to the 16S-1 rRNA gene in each sample using the 2^−ΔCT^ method, in which ΔC_T_ = C_T_ detected gene–C_T_ 16S-1 rRNA gene [19,48,50].

### 2.11. Statistical Analysis

Statistical analyses were performed using R Studio software (version 1.4.1103), R Markdown package [51]. The statistical analyses to compare the distribution of MIC values for antimicrobial agents and relative abundance of ARGs in *Aeromonas* strains isolated from healthy, furunculosis-suffering and antibiotic-treated fish and their environment on two fish farms were realised using the Kruskal–Wallis nonparametric ANOVA test. The statistical analyses to compare the occurrence of NWT Aeromonas strains and the presence of ARGs between the groups studied were realised using univariate analysis and binary logistic regression test for each antimicrobial agent or ARG. A significant difference was expressed as a *p*-value below the 5% confidence interval.

## 3. Results

### 3.1. Fish Farms Follow-Up and Clinical Observations

Two fish farms were surveyed for the presence of furunculosis outbreaks and antibiotic treatment from February to August 2020. On farm A, two episodes of furunculosis were confirmed by the veterinarian in May and July with mortality rates of around 2.1% and 3.4%, respectively. Bacteriological analysis showed the presence of dark-brown bacterial colonies typical of *Aeromonas salmonicida* on TSA agar (BIOKAR ref. BK047HA; Beauvais, France). Clinical signs, such as lesions on the skin, haemorrhagic intestinal tract and splenomegaly, were observed in sampled fish more in July than in May. In June, the mortality rate had decreased compared to May at 1.8% and no clinical signs were found in sampled fish. Fish were treated with flumequine at 12 g/kg feed for eight days in late July 2020 and almost no mortality was observed thereafter (0.1%). Therefore, two additional clinical samplings from moribund fish or fish with furuncle (boil or lesion) were realised on farm A in May and July. Therefore, the sampling in July was performed one day before starting the antibiotic treatment. Afterward, in August, one monthly sampling was carried out one week after the end of the antibiotic treatment. No episodes of furunculosis and no antibiotic treatments were observed on farm B.

### 3.2. Antimicrobial Susceptibility

A total of 257 *Aeromonas* spp. were selected for antimicrobial susceptibility tests from farms A and B, including 189 *Aeromonas* from fish samples and 68 from environmental strains (49 and 19 isolates from water and biofilm, respectively). Among these strains, 153 environmental and fish isolates were considered as healthy *Aeromonas* strains, including 58 and 98 strains isolated from farms A and B respectively, when no episode of furunculosis and no antibiotic treatment were observed. Fifty-four isolates were collected from fish with furunculosis signs after the confirmation of furunculosis on farm A. Fifty isolates were considered as treated environmental and fish isolates with the *Aeromonas* strains isolated following an antibiotic treatment.

For each antimicrobial susceptibility test, the MIC results obtained for the reference strains were in accordance with CLSI guidelines (data not shown) [39]. MIC value distributions of the seven antimicrobial agents and the corresponding MIC50 (median) and MIC90 (90th percentile) for 257 *Aeromonas* isolates are showed in Table 2. MIC values below the tested ranges varied but were all less than 12% for most of the antimicrobials tested, while MIC values above the tested ranges were not observed for any of the isolates and antimicrobials tested in this study. Differences between the MIC50 and MIC90 values were found for at least four dilutions for oxytetracycline (OXY), enrofloxacin (ENRO), florfenicol (FFN) and colistin (COL). Oxolinic acid (OA) and flumequine (FLUQ) showed five and six dilutions, respectively, and trimethoprim-sulfamethoxazole (TMP) presented the highest difference with seven dilutions between the MIC50 and MIC90 values.

COWTs were calculated for seven antimicrobial agents for all isolates using the Kronvall and/or Turnidge methods (Table 2). Similar COWTs were obtained by Kronvall and Turnidge methods for all antimicrobials except for OA and OXY. The difference in the COWTs was only one dilution for OA (Kronvall lower than Turnidge method) but five dilutions for OXY. For OA, due to its MIC values distribution, a COWT at 0.064 µg/mL using the Turnidge method seemed to be more appropriate to calculate the NWT isolates in this study. The COWT was computed to be 1 µg/mL for OXY using the Kronvall method, while it was calculated to be 32 µg/mL using the Turnidge method, which was greater than the highest MIC (16 µg/mL) for isolates tested in this study. Therefore, in this study, the COWT was considered at 1 µg/mL for OXY to calculate the number of non-wild-type (NWT) isolates or the isolates that presented MIC values higher than the COWTs.

The percentages of NWT *Aeromonas* ranged from 13% (FFN) to 60% (OXY). After oxytetracycline, the quinolone compounds (FLUQ, OA and ENRO) displayed the highest percentages, from 45 to 52% (Table 2). Among all of the isolates tested in this study, 109 (42%) isolates were considered as NWT strains for all three quinolone compounds (FLUQ, ENRO and OA).

### 3.3. Patterns of Antimicrobial Susceptibility in Aeromonas spp. Isolated from Healthy, Furunculosis-Suffering and Antibiotic-Treated Fish and Their Environment

The distributions of antimicrobial susceptibility of 257 *Aeromonas* strains isolated from healthy, furunculosis-suffering and FLUQ-antibiotic-treated fish and their environment for the antibiotics tested on fish farms A and B are presented in Figure 1. The MIC distributions appeared to have a similar pattern for the quinolone compounds, showing three distinct populations for FLUQ, ENRO and OA, while OXY, FFN, TMP and COL presented a bimodal pattern. For all isolates, MIC values were distributed in greater antimicrobial agent concentrations (more than the COWTs) on farm A than on farm B (*p* < 0.05), except for the FFN MIC values, which were distributed similarly on both farms, mostly less than the COWTs. In healthy *Aeromonas* strains, greater MIC values were observed only for OXY and OA on farm A compared to farm B (*p* < 0.05).

On farm A, no significant differences were observed in MIC values between the furunculosis *Aeromonas* strains and healthy isolates for all of the antibiotics tested (*p* > 0.05). Among the antibiotics tested, FLUQ, OA, ENRO (quinolone compounds), COL and TMP showed significantly higher MIC values for the FLUQ-treated isolates compared with the healthy *Aeromonas* strains on farm A (*p* < 0.05) (Figure 1).

### 3.4. Distribution of NWT Aeromonas and pMAR in Healthy, Furunculosis-Suffering and Antibiotic-Treated Fish and Their Environment

NWT *Aeromonas* isolates for each antibiotic tested in this study originated from different sample collections, namely, environmental (water and biofilm) samples and fish samples, with or without a furunculosis episode or flumequine treatment, on each farm (Figure 2). With regard to the occurrence of NWT *Aeromonas*, no significant differences were found between the environmental and fish samples studied on both farms for all antibiotics tested in healthy NWT *Aeromonas* (*p* > 0.05). Comparing the two farms, the occurrences of NWT *Aeromonas* for quinolone compounds (FLUQ, OA and ENRO), OXY and TMP were greater on farm A than on farm B for healthy strains (*p* < 0.05), but for FFN and COL, no significant differences were observed between farms A and B (*p* > 0.05). On farm A, furunculosis-suffering fish did not show a greater occurrence of NWT *Aeromonas* compared to healthy fish for all antimicrobial agents (*p* > 0.05). To study the presence of NWT *Aeromonas* for the healthy and FLUQ-treated strains studied, no differences were observed for OXY (healthy = 66% vs. FLUQ-treated = 68%). In contrast, the occurrence of treated NWT *Aeromonas* was greater than in healthy strains (*p* < 0.05) for COL (25% vs. 58%), FLUQ (48% vs. 76%), AO (52% vs. 74%), ENRO (57% vs. 78%), TMP (36% vs. 54%) and FFN (10% vs. 22%) (Figure 2). Nevertheless, these FLUQ-treated strains isolated from farm A were found with more NWT *Aeromonas* in the fish rather than in environmental samples for all antibiotics tested (*p* < 0.05).

In this study, approximately 36% (93 out of 257) isolates of *Aeromonas* strains were determined as being presumptive multi-antibiotic resistant bacteria (pMAR). With regard to the occurrence of pMAR *Aeromonas* for healthy strains, the presence of pMAR *Aeromonas* was greater on farm A than on farm B (37% vs. 24%) (*p* < 0.05). However, the distributions of pMAR *Aeromonas* for healthy isolates showed no significant differences between environmental and fish samples on both farms (*p* > 0.05).

On farm A, furunculosis-suffering fish did not show a greater occurrence of pMAR *Aeromonas* compared to healthy fish (*p* > 0.05). With regard to the presence of pMAR *Aeromonas* for the healthy and FLUQ-treated strains studied, a greater occurrence of treated pMAR *Aeromonas* rather than healthy ones was found (healthy = 32% vs. FLUQ-treated = 68%) (*p* < 0.05) and more pMAR *Aeromonas* were found in fish than in environmental samples (46% vs. 28%) (*p* < 0.05) (Figure 2).

### 3.5. Occurrence and Abundance of Aeromonas Antibiotic-Resistant Genes

Among the 257 *Aeromonas* spp., 211 isolates were selected for ARG analysis by considering their origin and antimicrobial susceptibility profiles. The occurrence, number of strains that express the gene and abundance estimated using the relative abundance (RA) of *Aeromonas* ARGs were studied for 44 specific genes including quinolones, tetracycline, sulfonamide-trimethoprim, phenicol, polymyxin, β-lactam and multidrug-resistance genes. Among these genes, 30 primers were detected and quantified in WT and NWT *Aeromonas* strains (Table 3). For ARGs involved in quinolone resistance, four genes were expressed: *qnr*A, *qnr*B, *aac6Ib*01 and *aac6Ib*02. The occurrence of *qnr*A and *aac6Ib*02 was significantly greater in WT than in NWT *Aeromonas,* but no differences were found in terms of the average RA (*p* > 0.05). For *qnr*B and *aac6Ib*01 genes, only one strain expressed these genes. This strain was an NWT *Aeromonas* isolated from a fish treated with flumequine from farm A, with a very high MIC for ENRO, FLUQ and OA (4, 8 and 32 µg/mL, respectively) and the highest abundance for *qnr*B, *aacb6Ib*01 and *aac6Ib*02 genes (0.655, 0.640 and 0.512, respectively).

Among the ARGs tested that are involved in tetracycline resistance, no differences between NWT and WT strains were observed for the occurrences and abundances of *tet*B2, *tet*C-02, *tet*D-02, *tet*G-02 and *tet*M1. For *tet*G01, although there were differences between the RAs of NWT and WT (*p* < 0.05), the RAs were very low and no differences in the occurrences were observed (*p* > 0.05). *Tet*E was the dominant antibiotic-resistant gene in 72% (152/211) of *Aeromonas* studied. The occurrence of *tet*E gene was significantly greater in NWT than in WT *Aeromonas* (123 and 29 respectively) (*p* < 0.05) but no significant differences were found for the RA between NWT and WT *Aeromonas* (0.06 and 0.02, respectively) (*p* > 0.05) (Table 3). Finally, significant differences in occurrence and abundance between NWT and WT strains were only observed for the tetA2 gene (*p* < 0.05).

Eight ARGs for sulfonamide-trimethoprim were expressed in *Aeromonas* strains. No differences between NWT and WT strains were observed for *str*A (occurrence and RA, *p* > 0.05). The occurrences, but not the RA, of *dfr*A1-1, *dfr*A1-2, *sul*1 and *str*B were significantly greater in NWT than in WT *Aeromonas* (*p* < 0.05). Conversely, the RA, but not the occurrence, of sul2 was significantly greater in NWT than in WT *Aeromonas* (*p* < 0.05). For *dfr*A12 gene, only six NWT strains expressed this gene. These six strains were all isolated on farm A from fish treated with flumequine and had high MIC (2-38/32-608 µg/mL for TMP). The occurrence and RA were significantly greater for NWT than WT strains only for sul3 (*p* < 0.05) (Table 3). For florfenicol, *floR*-1 was detected with a higher occurrence and RA for NWT *Aeromonas* than for WT strains (*p* > 0.05). Finally, for the polymyxin family, only *mcr*2 and *mcr*3 genes were expressed but at a very low level and there were no differences in their occurrence or RA between the NWT and WT strains (not applicable and *p* > 0.05, respectively) (Table 3).

Among the ARGs tested, *tet*A2, *sul*3 and *floR*1 were detected with a higher significant occurrence and abundance in NWT than in WT *Aeromonas* (*p* < 0.05). The distribution of these ARGs in WT and NWT *Aeromonas* is displayed for healthy, furunculosis-suffering and antibiotic-treated fish and their environment on the two fish farms (Figure 3). To compare the occurences of ARGs in “healthy” *Aeromonas* isolated from the two farms (fish and environment), no significant differences were found for the three ARGs (*p* > 0.05). Similarly, no significant differences were found between healthy environmental and fish strains isolated from both farms (*p* > 0.05). On farm A, more *tet*A2, but not *sul*3 and *floR*1 genes, were detected for strains isolated from furunculosis fish compared to healthy ones. For the three ARGs, no significant differences were seen between FLUQ-treated *Aeromonas* and healthy fish and environmental strains (*p* > 0.05) (Figure 3).

MexF genes were observed in 136 out of 211 studied isolates, including 67 and 69 pMAR and non-pMAR strains, respectively. The occurrence and abundance of *mex*F were compared between these strains. This gene showed a higher average RA in pMAR *Aeromonas* than in non-pMAR *Aeromonas* (0.069 vs. 0.005, *p* < 0.05). Similarly, the occurrences of *mex*F genes were significantly greater in pMAR rather than in non-pMAR *Aeromonas* (67/91 (73%) vs. 69/120 (57%), respectively) (*p* < 0.05). No significant differences were found for *mex*F in healthy *Aeromonas* strains between the two farms (*p* > 0.05) (Figure 3). However, more *mex*F genes were found in fish samples than in environmental samples on farm A (*p* < 0.05), while on farm B, the occurrence of *mex*F genes in water and biofilm samples was higher than in fish samples (*p* < 0.05) in healthy *Aeromonas*. On farm A, no significant differences were seen between furunculosis and healthy strains for *mex*F (*p* > 0.05). *Mex*F genes were more present in FLUQ-treated *Aeromonas* than in healthy isolates from fish samples, but not in environmental samples (*p* < 0.05) (Figure 3).

Three ARGs for β-lactams were expressed in *Aeromonas* strains, including *bla*SHV-01, *bla*-IMP2 and *bla*-KPC3 genes. *Bla*-IMP2 and *bla*-KPC3 genes were expressed but at very low occurrences and abundances. In contrast, *bla*SHV-01 genes were observed in 144 out of 211 *Aeromonas* strains with an RA at 0.0056. To analyse the occurrence of *bla*SHV-01 genes in “healthy” *Aeromonas* isolated from the two farms (fish and environment), no significant differences were found between the two farms (*p* > 0.05). However, more blaSHV-01 genes were found in fish samples than in environmental samples on farm A (*p* < 0.05), while on farm B, the occurrence of *blaSHV*-01 was higher in water and biofilm samples than in fish samples (*p* < 0.05) in healthy *Aeromonas*. On farm A, more *bla*SHV-01 genes were detected for strains isolated from furunculosis fish compared to healthy fish (*p* < 0.05). A higher presence of *bla*SHV-01 in FLUQ-treated *Aeromonas* than in healthy ones was observed in both fish and environmental strains (*p* < 0.05) (Figure 3).

## 4. Discussion

In this study, MIC distributions of 257 *Aeromonas* strains isolated from fish, water and biofilms on two rainbow trout farms over seven months were determined for antibiotics that are commonly used against *Aeromonas* infections. MIC distributions and the MIC50 and MIC90 values calculated showed a few differences compared to a previous study [31], with our study finding three distinct populations for quinolones (FLUQ: <0.125, 0.25–2, >4 µg/mL; OA: <0.032, 0.064–1, >2 µg/mL; ENRO: <0.032, 0.064–0.25, >0.5 µg/mL) and much higher values for quinolones and OXY. The greatest difference was found in the MIC50 value for TMP (0.03/0.6 µg/mL) which was lower than those observed in four different studies relative to *Aeromonas* [31,40,52,53]. These differences can be explained by differences between *Aeromonas* species, in the sources (environmental or fish) and locations (isolated or farming area) where the strains were isolated and in the occurrences of diseases with antibiotic treatments.

Aside from the methods used to calculate the COWTs (Kronvall or Turnidge) and a few differences in the COWTs for some antimicrobial agents, such as OA and OXY, our results were in accordance with the results obtained by Baron et al. [31] and Duman et al. [30], although the origins of the *Aeromonas* strains were different between these two studies (freshwater of different rivers or cultured fish) and our study (water, biofilm and fish samples from fish farms that included an episode of furunculosis and FLUQ treatment).

Few epidemiological cut-off values for *Aeromonas* could be found in the reference reports of antimicrobial susceptibility testing. CLSI has proposed the epidemiological cut-off values of *Aeromonas salmonicida* for FFN (4 µg/mL) and ormetoprim-sulfadimethoxine (0.5/9.5 µg/mL), and the clinical breakpoint for OXY (*susceptible:* ≤1 µg/mL) and OA (*susceptible:* ≤0.12 µg/mL) [54]. In addition, EUCAST has determined the clinical breakpoint of *Aeromonas* spp. for TMP (*susceptible: ≤* 2 µg/mL) [32]. Aside from *Aeromonas* species, our COWTs were close to these values indicated by CLSI and EUCAST. It was argued that in the absence of a clinical breakpoint for various antimicrobial agents, especially in *Aeromonas* spp., epidemiological cut-off values could be used to detect and monitor resistance [55]. Although interpretative criteria change over time, determining COWT and delineating WT (susceptible) from NWT (not susceptible) populations allowed us to evaluate antibiotic resistance profiles in *Aeromonas* spp.

On both farms A and B, NWT and pMAR isolates were detected in healthy *Aeromonas* strains from fish and environment samples, but the occurrence of NWT for FLUQ, OA, ENRO, OXY, TMP and pMAR *Aeromonas* was higher on farm A than farm B. The detection of antibiotic-resistant and MAR *Aeromonas* spp. on rainbow trout farms and in other various freshwater environments was previously reported by several authors, revealing that the presence of ARB could be due to the history of diverse antibiotic administrations on fish farms and/or to various animal and human activities in upstream areas [9,28,29]. On both farms studied, to our knowledge, antibiotic treatments were prescribed two, three and four years previously. The higher presence of NWT and pMAR *Aeromonas* strains on farm A may be due to the input river water being contaminated by various human activities and by effluents from the other fish, pig and cattle breeding sites located upstream of farm A, while farm B was situated in an isolated area. Naviner et al. [9] observed *Aeromonas* quinolone-resistant strains prior to an antibiotic treatment on a trout farm where the water was contaminated by effluents of farm activities upstream of the fish farm.

One week after the FLUQ treatment, we found that the occurrence of NWT for quinolones (FLUQ, AO and ENRO) and also for other antimicrobial classes, such as COL, TMP and FFN, as well as pMAR *Aeromonas* in FLUQ-treated isolates, was greater than in healthy isolates. Similarly, Naviner et al. [9] presented more quinolone-resistant *Aeromonas* strains after FLUQ treatment compared to prior antibiotic exposure on a rainbow trout farm. They also presented the resistance profiles of other antimicrobial classes, such as OXY, TMP and FFN, in FLUQ-treated isolates. The increase of *Aeromonas* spp. resistant to quinolones and other antimicrobial classes may be associated with FLUQ treatment for which genetic determinants responsible for the resistance are frequently carried on mobile genetic elements, such as plasmids, transposons and integrons borne on specific transposons or plasmids [9,56,57]. The occurrence of NWT and pMAR *Aeromonas* increased quickly (one week in our study) and then could persist for at least 22 days after the FLUQ treatment on the fish farm [9]. Similarly, Guardabassi et al. [58] already found the persistence of antibiotic-resistant *Acinetobacter* in the water of a trout farm up to six months after the end of the OA treatment.

A high presence of NWT *Aeromonas* for OXY (67% of *Aeromonas*) was observed in this study, which could be explained mainly by the predominant occurrence of the *tet*E efflux pump gene in NWT rather than in WT *Aeromonas* (COWT: 1 µg/mL; 123 vs. 29 isolates). However, the gene *tet*A2 showed significant differences between the NWT and WT strains for both occurrence and abundance. Previous research showed the high occurrence of the *tet*E gene in NWT *Aeromonas* (43 vs. 22 isolates; COWT: 2 µg/mL) [30]. Similarly, *tet*E and/or *tet*A were detected as a common *tet* gene studied (A–E) in motile *Aeromonas* strains from Danish and Turkish fish farms and environments [30,59]. In our study, the occurrence of efflux pump genes (*tet*A–G) was greater than ribosomal protection protein *tet* gene (*tet*M) in *Aeromonas*. However, Muziasari et al. [19] found high abundances of *tet*M in intestinal DNA from farm-raised salmonid fish.

Although 105 out 211 *Aeromonas* strains (49%) were considered as NWT for the three quinolones (FLUQ, OA and ENRO), all four plasmid-mediated quinolone-resistance genes studied, including *qnr* (A and B) and *aac6Ib* (01 and 02) genes, did not seem to be involved in this quinolone resistance. Indeed, the occurrence of *qnr*A and *aac6Ib*02 was significantly greater in WT than in NWT *Aeromonas* (no differences were found between the average RAs). For the *qnr*B and *aac6Ib*01 genes, only one strain expressed these genes. This may be explained by the potential presence of other quinolone ARGs that have not been studied, such as *qnr*S and *aac-6′-Ib-cr* [60,61]. Our findings are in accordance with a previous study in which neither the *qnr*A nor the *qnr*B gene was detected in any of the 40 resistant *Aeromonas hydrophila* strains isolated from aquatic animals, and only two strains were detected with *aac6Ib,* while all the enrofloxacin-resistant isolates harbored *qnr*S plasmid-mediated quinolone-resistance genes [60]. Conversely, Chenia [61] showed no *aac-6′-Ib-cr* but a high prevalence of *qnr*B and *qnr*S (41% and 24% respectively) for *Aeromonas* spp. isolated from South African freshwater fish. However, in our study, a highly resistant strain to the three tested quinolones also expressed the highest abundance for *qnr*B and *aac6Ib* (01 and 02) genes, showing the importance of these genes in the resistance to quinolones.

Around 38% of *Aeromonas* spp. (91 out of 211 isolates) were determined to be NWT isolates for TMP, which can be linked to the presence of eight studied ARGs, mainly sulfonamide-resistance genes, such as *sul*1, *sul*2, *sul*3 and *str*B, and trimethoprim-resistance genes (dihydrofolate reductase), such as *dfr*A1-1 and *dfr*A1-2, which were expressed significantly more in NWT than in WT *Aeromonas* (occurrence or abundance). However, only *sul*3 showed a higher occurrence and RA in NWT than in WT strains. Of the TMP-resistance genes studied, the *sul*3 gene may therefore play a greater role in the spread of resistant *Aeromonas* bacteria in aquatic environments. Duman et al. [30] and Capkin et al. [62] reported *sul*1 as the most common TMP resistance gene in *Aeromonas* species, but Piotrowska and Popowska [10] indicated a higher presence of *sul*2. The differences observed between studies can be attributed to the regional diversity of the isolates.

Although 78 out of 211 strains (36%) were determined as NWT for colistin, only a few polymyxin genes, such as *mcr*2 and *mcr*3 genes (1 and 4 strains, respectively), were detected, while *mcr*1–5 were the resistance genes most found among *Aeromonas* species and other Gram-negative bacteria, such as *E. coli* [63]. In our study, resistance to florfenicol (COWT: 2 µg/mL) was found for 31 out 211 isolates (14%), which can be associated with the higher occurrence and abundance of the *floR*-1 efflux pump gene in NWT than in WT isolates. Our results are in line with previous authors who considered that most pathogenic bacteria in fish, including *Aeromonas* spp., mediate florfenicol resistance through *floR* [30,64]. Although β-lactam antibiotics are not used in aquaculture, *bla*SHV-01 was detected in 144 out of 211 (68%) *Aeromonas* strains in our study. Indeed, *Aeromonas* strains seem to be intrinsically resistant to this antibiotic family [65]. Two previous studies showed the low sensitivity of *Aeromonas* strains to β-lactams and an unexpected imipenem and the presence of *bla*CphA/IMIS *intI*1 and *bla*SHV (ESBL genes with class 1 integron) in *Aeromonas* from farmed rainbow trout [28,29]. Therefore, resistance to β-lactams in ubiquitous *Aeromonas* bacteria can be a great concern for public health due to the frequent administration of these antibiotics in human medicine [28].

The mex systems were associated with multidrug resistance genes, such as *Mex*AB-OprM, *Mex*CD-OprJ and *Mex*EF-OprN. In our study, 136 of the 211 *Aeromonas* studied carried *mex*F genes, with the occurrence and RA greater in pMAR *Aeromonas* than in non-pMAR *Aeromonas* for five antimicrobial classes. To our knowledge, it is the first description of a mex system detected for *Aeromonas spp*. Only *AheABC* multidrug efflux pump was expressed in *A. hydrophila* at a low level involving an intrinsic multidrug resistance [66].

By comparing the *bla*SHV-01 and *mex*F distributions on the two fish farms, we found the same profiles of antimicrobial resistance among *Aeromonas*. The occurrence of these genes was significantly higher in fish than in water and biofilm collected from farm A, while farm B showed the inverse. This may be explained by the different location of each farm and their distance from other animal and human facilities, which may have resulted in the spread of ARB and ARGs. As farm B was situated in an isolated area, the ARGs could have come from a long distance away. Previous studies found that antibiotic-resistant bacteria and resistance genes may be transferred by the water current and persist even over a long distance (20 km downstream) [67].

Furthermore, a greater occurrence of pMAR and quinolone-resistant bacteria on the one hand, and *bla*SHV-01 and *mex*F genes on the other, were detected in *Aeromonas* spp. isolated from FLUQ-treated fish and their environment than in healthy strains. Previous findings revealed that a two-component regulatory system of two proteins (an inner membrane histidine kinase and a cytoplasmic response regulator) interconnects resistance to polymyxins, (fluoro)quinolones and β-lactams in *Pseudomonas aeruginosa*. The mechanisms of resistance for these antimicrobial agents, such as an altered permeability, increased drug efflux and reduced porin pathway of the bacterial membrane could be integrated through an overexpression of the *mex* efflux system in Gram-negative bacteria, such as *Pseudomonas aeruginosa* [68]. Similarly, as Gram-negative bacteria, *Aeromonas* spp. might harbour the multidrug resistance mechanisms for quinolone and β-lactam antimicrobial agents mainly after a FLUQ treatment.

## 5. Conclusions

This study demonstrated that aquaculture farms may be considered as a huge environmental reservoir of multidrug-resistant bacteria and ARGs, and the results suggest that *Aeromonas* may be used as an indicator of antimicrobial susceptibility for aquatic ecosystems. Our findings clearly show that human and animal husbandry activities on the one hand, and antibiotic treatments administered on fish farms on the other, impact the presence and dissemination of ARB and ARGs in fish and their environment. There is thus a high risk that resistance genes may develop and spread between fish, their environments and humans. Future research should focus on screening and quantifying plasmids and other mobile genetic elements involved in antimicrobial resistance from *Aeromonas* isolates in aquatic systems and their persistence in the environment also should be studied. Moreover, the maintenance and dissemination of ARB and ARGs associated with antibiotics that are mainly applied in aquaculture and also are used in human medicine need to be examined. Our findings point out that the increase and persistence of ARB and ARGs on mobile genetic elements after an antibiotic treatment on a fish farm might have a great impact on human, animal and environmental health. Furthermore, sustainable aquaculture practices investing in new approaches to reduce the spread of antibiotic resistance need to be established.

## Figures and Tables

**Figure 1 microorganisms-09-01201-f001:**
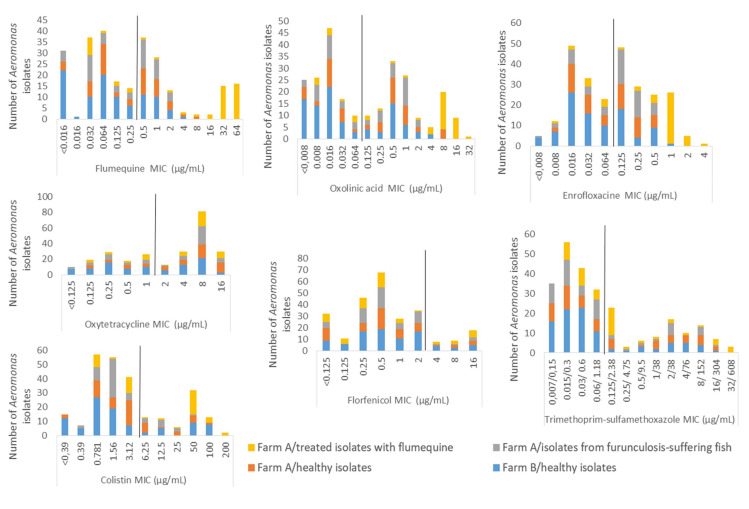
The distributions of antimicrobial susceptibility of 257 *Aeromonas* strains isolated from healthy, furunculosis-suffering and flumequine-antibiotic-treated fish and their environment for antibiotics tested on two fish farms (A and B). The calculated COWTs are shown with a bar to define the wild-type (before the bar) and non-wild-type (after the bar) populations in this study.

**Figure 2 microorganisms-09-01201-f002:**
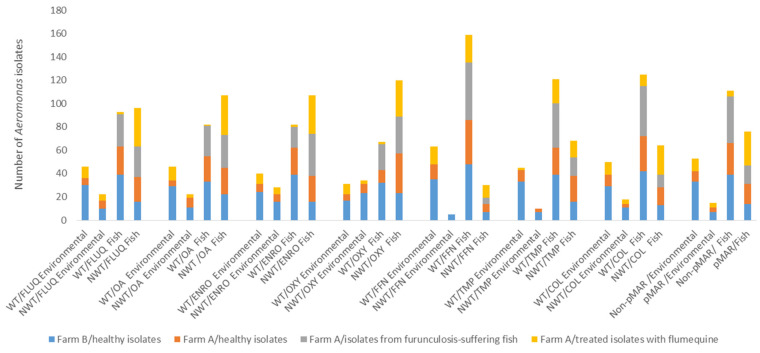
The distribution of wild-type (WT) and non-wild-type (NWT) *Aeromonas* isolated from environmental (water pond and biofilm) and fish samples for each antimicrobial agent on two fish farms (A and B). *Aeromonas* strains isolated from healthy, furunculosis-suffering and antibiotic-treated fish and their environment. Note: no episode of furunculosis or antibiotic treatment was observed on farm B; FLUQ: flumequine; OA: oxolinic acid; ENRO: enrofloxacin; OXY: oxytetracycline; FFN: florfenicol; TMP: trimethoprim-sulfamethoxazole; COL: colistin; pMAR: presumptive multi-antibiotic-resistant *Aeromonas*.

**Figure 3 microorganisms-09-01201-f003:**
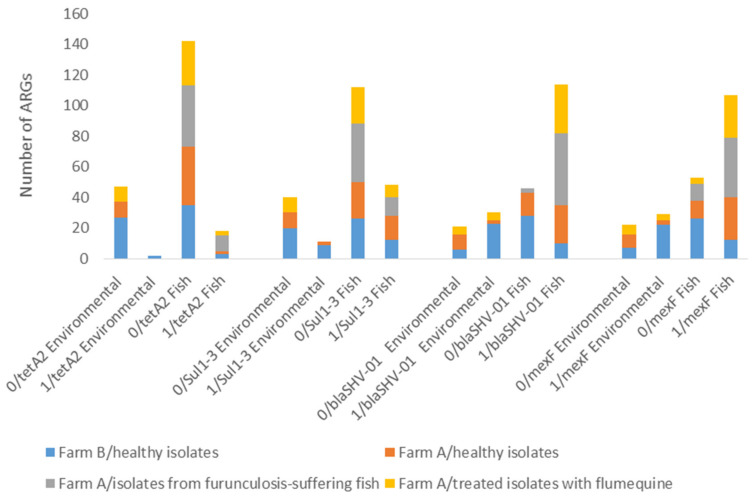
The distribution of antibiotic resistance genes (ARGs) in *Aeromonas* isolated from environmental and fish samples on fish farms A and B. *Aeromonas* strains isolated from healthy, furunculosis-suffering and antibiotic-treated fish and their environment. Note: no episode of furunculosis or antibiotic treatment was observed on farm B; 0: ARG has not been detected; 1: ARG has been detected.

**Table 1 microorganisms-09-01201-t001:** Primers used for PCR amplification. F: forward, R: reverse.

Primer Name			Amplicon Size (bp)
	F	R	
	Sequence (5′→3′)		
***qnr*** **A**	AGGATTTCTCACGCCAGGATT	CCGCTTTCAATGAAACTGCAA	123
***qnr*** **B**	GCGACGTTCAGTGGTTCAGA	GCTGCTCGCCAGTCGAA	61
***aac(6′)-Ib*** **-01**	GTTTGAGAGGCAAGGTACCGTAA	GAATGCCTGGCGTGTTTGA	72
***aac(6′)-Ib*** **-02**	CGTCGCCGAGCAACTTG	CGGTACCTTGCCTCTCAAACC	65
***dfrA1*** **-01**	GGAATGGCCCTGATATTCCA	AGTCTTGCGTCCAACCAACAG	94
***dfrA1-*** **02**	TTCAGGTGGTGGGGAGATATAC	TTAGAGGCGAAGTCTTGGGTAA	149
***dfrA*** **12**	CCTCTACCGAACCGTCACACA	GCGACAGCGTTGAAACAACTAC	84
***sul*** **1**	CAGCGCTATGCGCTCAAG	ATCCCGCTGCGCTGAGT	128
***sul*** **2**	TCCGATGGAGGCCGGTATCTGG	CGGGAATGCCATCTGCCTTGAG	101
***sul*** **3**	GCCGATGAGATCAGACGTATTG	CGCATAGCGCTGGGTTTC	189
***str*** **A**	AATGAGTTTTGGAGTGTCTCAACGTA	AATCAAAACCCCTATTAAAGCCAAT	147
***str*** **B**	GCTCGGTCGTGAGAACAATCT	CAATTTCGGTCGCCTGGTAGT	100
***mcr*** **-1**	CGGTCAGTCCGTTTGTTC	CTTGGTCGGTCTGTAGGG	308
***mcr*** **-2**	TGTTGCTTGTGCCGATTGGA	AGATGGTATTGTTGGTTGCTG	562
***mcr*** **-3**	TTGGCACTGTATTTTGCATTT	TTAACGAAATTGGCTGGAACA	542
***mcr*** **-4**	ATTGGGATAGTCGCCTTTTT	TTACAGCCAGAATCATTATCA	487
***mcr*** **-5**	ATGCGGTTGTCTGCATTTATC	TCATTGTGGTTGTCCTTTTCTG	1644
***tet*** **A-01**	GCTGTTTGTTCTGCCGGAAA	GGTTAAGTTCCTTGAACGCAAACT	62
***tet*** **A-02**	CTCACCAGCCTGACCTCGAT	CACGTTGTTATAGAAGCCGCATAG	100
***tet*** **B-01**	AGTGCGCTTTGGATGCTGTA	AGCCCCAGTAGCTCCTGTGA	62
***tet*** **B-02**	GCCCAGTGCTGTTGTTGTCAT	TGAAAGCAAACGGCCTAAATACA	100
***tet*** **C-01**	CATATCGCAATACATGCGAAAAA	AAAGCCGCGGTAAATAGCAA	77
***tet*** **C-02**	ACTGGTAAGGTAAACGCCATTGTC	ATGCATAAACCAGCCATTGAGTAAG	104
***tet*** **D-01**	TGCCGCGTTTGATTACACA	CACCAGTGATCCCGGAGATAA	85
***tet*** **D-02**	TGTCATCGCGCTGGTGATT	CATCCGCTTCCGGGAGAT	100
***tet*** **E**	TTGGCGCTGTATGCAATGAT	CGACGACCTATGCGATCTGA	73
***tet*** **G-01**	TCAACCATTGCCGATTCGA	TGGCCCGGCAATCATG	92
***tet*** **G-02**	CATCAGCGCCGGTCTTATG	CCCCATGTAGCCGAACCA	139
***tet*** **M-01**	CATCATAGACACGCCAGGACATAT	CGCCATCTTTTGCAGAAATCA	100
***tet*** **M-02**	TAATATTGGAGTTTTAGCTCATGTTGATG	CCTCTCTGACGTTCTAAAAGCGTATTAT	146
***tet*** **M-03**	GCAATTCTACTGATTTCTGC	CTGTTTGATTACAATTTCCGC	185
***floR*** **-01**	ATTGTCTTCACGGTGTCCGTTA	CCGCGATGTCGTCGAACT	60
***cat*** **A1**	GGGTGAGTTTCACCAGTTTTGATT	CACCTTGTCGCCTTGCGTATA	100
***bla*** **ACC**	CACACAGCTGATGGCTTATCTAAAA	AATAAACGCGATGGGTTCCA	67
***bla*** **CMY**	CCGCGGCGAAATTAAGC	GCCACTGTTTGCCTGTCAGTT	107
***bla*** **CTX-M-01**	GGAGGCGTGACGGCTTTT	TTCAGTGCGATCCAGACGAA	91
***bla*** **DHA**	TGGCCGCAGCAGAAAGA	CCGTTTTATGCACCCAGGAA	120
***bla*** **IMP-01**	AACACGGTTTGGTGGTTCTTGTA	GCGCTCCACAAACCAATTG	100
***bla*** **IMP-02**	AAGGCAGCATTTCCTCTCATTTT	GGATAGATCGAGAATTAAGCCACTCT	232
***bla*** **IMP-03**	GGAATAGAGTGGCTTAATTC	GGTTTAACAAAACAACCACC	71
***bla*** **KPC-02**	CAGCTCATTCAAGGGCTTTC	GGCGGCGTTATCACTGTATT	195
***bla*** **KPC-03**	GCCGCCGTGCAATACAGT	GCCGCCCAACTCCTTCA	59
***bla*** **SHV-01**	TCCCATGATGAGCACCTTTAAA	TTCGTCACCGGCATCCA	69
***mex*** **F**	CCGCGAGAAGGCCAAGA	TTGAGTTCGGCGGTGATGA	287
***16S*** **-01**	GGGTTGCGCTCGTTGC	ATGGYTGTCGTCAGCTCGTG	60
***16S*** **-02**	CCTACGGGAGGCAGCAG	ATTACCGCGGCTGCTGGC	195
***rpo*** **B**	CGAACATCGGTCTGATCAACTC	GTTGCATGTTCGCACCCAT	359

**Table 2 microorganisms-09-01201-t002:** Distribution of MIC values (µg/mL) in 257 isolates of *Aeromonas* spp.

MIC (µg/mL)	0.008	0.016	0.032	0.064	0.125	0.25	0.5	1	2	4	8	16	32	64	128	256	HR (%)	MIC50	MIC90	COWT Kronvall (K) and Turnidge (T)	NWT (%)
Flumequine		1	37	40	16	14	39	28	13	3	2	2	15	16			31 (12%)	0. 5	32	0.25 (K,T)	118 (45%)
Oxolinic acid	29	47	17	10	10	13	33	27	9	5	22	9	1				25 (9%)	0.25	8	0.032 (K) 0.064 (T)	139 (53%) (T)
Enrofloxacin	12	49	33	23	48	29	28	24	5	1							5 (2%)	0.125	1	0.064 (K,T)	135 (52%)
Oxytetracycline					19	29	18	26	13	30	82	30					10 (3%)	4	16	1 (K) 32 (T)	155 (60%) (K)
Florfenicol					11	50	68	28	35	8	9	18					32 (12%)	0.5	8	2 (K,T)	35 (13%)
**MIC (µg/mL)**	**0.007/0.15**	**0.015/0.3**	**0.03/0.6**	**0.06/1.18**	**0.125/2.38**	**0.25/4.75**	**0.5/9.5**	**1/19**	**2/38**	**4/76**	**8/152**	**16/304**	**32/608**				**HR (%)**	**MIC50**	**MIC90**	**Kronvall (K) and** **Turnidge (T)**	**NWT (%)**
Trimethoprim-Sulfamethoxazole	35	56	43	32	23	3	6	8	15	12	14	7	3				-	0.03/0.6	4/76	0.06/1.18 (K,T)	91 (35%)
**MIC (µg/mL)**	**0.39**	**0.781**	**1.56**	**3.12**	**6.25**	**12.5**	**25**	**50**	**100**	**200**							**HR (%)**	**MIC50**	**MIC90**	**Kronvall (K) and** **Turnidge (T)**	**NWT (%)**
Colistin	7	57	56	41	13	12	6	32	14	**5**							15 (5%)	3.12	50	3.12 (K,T)	83 (32%)

Note: Gray color represents the selected range of dilutions for the MIC values study. HR: number (percentage) of isolates for which the MIC value was below the test range (no isolate had an MIC value above the test range) in this study. COWTs: epidemiological cut-off values were calculated using two methods, namely, Kronvall (K) and Turnidge (T). Green color represents MIC > COWT or the isolates of non-wild-type (NWT) resulting from the Kronvall and/or Turnidge method. NWT (%): number (percentage) of isolates for which the MIC values were above the COWT considered in this study.

**Table 3 microorganisms-09-01201-t003:** The relative abundance (RA) (relative to the 16S rRNA gene) and distribution of antibiotic resistance genes (ARGs) in wild-type (WT) and non-wild-type (NWT) Aeromonas isolated from healthy, furunculosis-suffering and antibiotic-treated fish and their environment on two fish farms. FLUQ: flumequine; OA: oxolinic acid; ENRO: enrofloxacin; OXY: oxytetracycline; FFN: florfenicol; TMP: trimethoprim-sulfamethoxazole; COL: colistin; pMAR: presumptive multi-antibiotic-resistant Aeromonas; *NS*: *not significant*; *NA*: *not applicable*.

	*Aeromonas* spp. (n = 211)		ARGs	NWT			WT				
	**NWT (n)**	**WT (n)**		**n**	**Mean (RA)**	**SD**	**n**	**Mean (RA)**	**SD**	***p*-Value** **(n)**	***p*-Value** **(RA)**
3 Quinolones	105	106	*qnr*A	34	0,00016	0.00027	47	0.00011	0.00014	*p* < 0.05	*NS*
FLUQ	113	98	*qnr*B	1	0.65586		0			*NA*	*NA*
OA	123	88	*aac(6′)-Ib*-01	1	0.64029		0			*NA*	*NA*
ENRO	128	83	*aac(6′)-Ib*-02	11	0.03037	0.12418	20	0.00018	0.00016	*p* < 0.05	*NS*
OXY	143	68	*tet*A2	17	0.15126	0.30516	5	0.00013	0.00010	*p* < 0.05	*p* < 0.05
			*tet*B2	74	0.00017	0.00029	40	0.00019	0.00025	*NS*	*NS*
			*tet*C-02	27	0.00048	0.00045	10	0.00087	0.00092	*NS*	*NS*
			*tet*D-01	6	0.00015	0.00010	0			*NA*	*NA*
			*tet*D-02	2	0.00003	0.00001	1	0.00019	0.00019	*NS*	*NS*
			*tet*E	123	0.05969	0.09113	29	0.02540	0.05679	*p* < 0.05	*NS*
			*tet*G-01	38	0.00039	0.00046	21	0.00077	0.00069	*NS*	*p* < 0.05
			*tet*G-02	9	0.00007	0.00008	4	0.00006	0.00004	*NS*	*NS*
			*tet*M1	13	0.00116	0.00381	7	0.00007	0.00002	*NS*	*NS*
			*tet*M2	1	0.01930		0			*NA*	*NA*
			*tet*M3	1	0.00031		0			*NA*	*NA*
TMP	81	130	*sul*1	41	0.05938	0.02253	5	0.02947	0.04186	*p* < 0.05	*NS*
			*sul*2	26	0.03711	0.07584	32	0.00046	0.00114	*NS*	*p* < 0.05
			*sul*3	45	0.13263	0.05046	14	0.02422	0.06013	*p* < 0.05	*p* < 0.05
			*dfr*A1-1	25	0.16801	0.06140	3	0.13359	0.11725	*p* < 0.05	*NS*
			*dfr*A1-2	18	0.12146	0.05408	4	0.05189	0.06499	*p* < 0.05	*NS*
			*dfr*A12	6	0.06721	0.00377	0			*NA*	*NA*
			*str*A	32	0.00042	0.00072	45	0.00023	0.00049	*NS*	*NS*
			*str*B	16	0.16930	0.23542	2	0.07267	0.12576	*p* < 0.05	*NS*
FFN	31	180	*floR*-1	16	0.06315	0.04174	24	0.00375	0.01542	*p* < 0.05	*p* < 0.05
COL	78	133	*mcr*2	0			1	0.00020		*NA*	*NA*
			*mcr*3	1	0.00004		3	0.01849	0.01841	*NS*	*NS*
			**ARGs**	**n**	**Mean(RA)**	**SD**					
β-lactams	-	-	*bla*SHV-01	144	0.0056534	0.0104554					
			*bla*-KPC3	8	5.45 × 10^−5^	4.6 × 10^−5^					
			*bla*-IMP2	1	6.68 × 10^−5^	NA					
			**ARGs**		**pMAR**			**Non-pMAR**			
	**pMAR (n)**	**Non-pMAR (n)**		**n**	**Mean (RA)**	**SD**	**n**	**Mean (RA)**	**SD**		***p*-value**
Multi-resistant	91	120	*mex*F	67	0.0699265	0.0706432	69	0.0055260	0.0089965	*p* < 0.05	*p* < 0.05

## Data Availability

Data sharing is not applicable; no new data was generated.

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
