# Peer review of "Antimicrobial Susceptibility Profiles and Resistance Genes in Genus *Aeromonas* spp. Isolated from the Environment and Rainbow Trout of Two Fish Farms in France"

_microorganisms, 2021, doi:10.3390/microorganisms9061201_

Round 1

Reviewer 1 Report

The theme of this study is greatly relevant in present times where the concern over antimicrobial resistance among pathogenic bacteria is rising.

It is very surprising that there are no line numbers in the manuscript, which makes extremely hard to put the comments on the manuscript.

Comments:

  1. What was the basis for the statistical tests used Kruskal-Wallis and logistic regression?
  2. What is the meaning of healthy Aeromonas strains?
  3. Please add water temperature of fish farms.
  4. What was the size and weight of sampled fish?
  5. Table 1: add the size of each amplified primer set.
  6. The quality of Figure 1, 2 and 3 is very poor. Please provide high quality of Figures.
  7. What are the three distinct populations for quinolones which the current study claims to find? From section 4.Discussion MIC distributions and the MIC50 and MIC90 values calculated showed a few differences compared to a previous study [31], with our study finding three distinct populations for quinolones and much higher values for quinolones and OXY.
  1. Language of text may be simplified to provide better clarity for the readers. As an example, in section 2.6 Biofilms samples “One biofilm surface was taken for the bacteriological analysis of cumulative previous months, and another biofilm surface was also collected from a previous month and then replaced with the biofilm surface of the following month. Each biofilm plate was placed in a sterile bottle filled with the corresponding pond water “.
  2. Few grammatical errors were noticed. As an example In lines from section 4.Discussion “Although 105 out 211 Aeromonas strains (49%) were consider as NWT for the three quinolones (FLUQ, OA and ENRO)”.

Author Response

Dear reviewer,

Thank you for your letter and the opportunity to revise our paper. The suggestions offered have been immensely helpful, and we also appreciate your insightful comments on revising the manuscript and other aspects of the paper.

I have included your comments immediately after this letter and responded to them individually indicating exactly how we addressed each concern or problem and describing the changes we have made. The changes are inserted and also marked in “Track Changes” in the revised paper. The revised manuscript is also submitted online by using the Editorial Manager system.

Comments:

  • It is very surprising that there are no line numbers in the manuscript, which makes extremely hard to put the comments on the manuscript:

Response: We are very sorry for this inconvenience but we have uploaded the manuscript (PDF) with line numbers.

  1. What was the basis for the statistical tests used Kruskal-Wallis and logistic regression?

Response:

Kruskal-Wallis: Nonparametric ANOVA 

logistic regression: Univariate analysis and binary logistic regression

  1. What is the meaning of healthy Aeromonas strains?

Response: the sentences are modified and added.

In “Aeromonas spp. isolation and identification” part:

These isolates were classified as healthy isolates when no episode of furunculosis or no antibiotic treatment were observed; furunculosis isolates when furunculosis occurred; treated isolates when followed by an antibiotic treatment.

In “Antimicrobial susceptibility” part:

Among these strains, 153 environmental and fish isolates were considered as healthy Aeromonas strains including 58 and 98 strains isolated from farms A and B respectively, when no episode of furunculosis and no antibiotic treatment have been observed.

  1. Please add water temperature of fish farms:

Response:  The average water temperature on both fish farms is recorded at 10± 0.5 °C and 14± 0.5 °C in winter and summer season respectively.

  1. What was the size and weight of sampled fish?

 Response: These fish farms are composed of upstream ponds for juvenile trout and downstream ponds for larger trout until they reach the commercial weight. In this study, the large trout from 40± 5 cm/ 800± 200 g to 55± 5 cm/ 2000± 200 g were considered at the beginning and end of the study respectively.

  1. Table 1: add the size of each amplified primer set: Response: Table1 is modified.

Primer Name

Amplicon Size (bp)

F

R

Sequence (5’ à 3’)

qnrA

AGGATTTCTCACGCCAGGATT

CCGCTTTCAATGAAACTGCAA

123

qnrB

GCGACGTTCAGTGGTTCAGA

GCTGCTCGCCAGTCGAA

61

aac(6')-Ib-01

GTTTGAGAGGCAAGGTACCGTAA

GAATGCCTGGCGTGTTTGA

72

aac(6')-Ib-02

CGTCGCCGAGCAACTTG

CGGTACCTTGCCTCTCAAACC

65

dfrA1-01

GGAATGGCCCTGATATTCCA

AGTCTTGCGTCCAACCAACAG

94

dfrA1-02

TTCAGGTGGTGGGGAGATATAC

TTAGAGGCGAAGTCTTGGGTAA

149

dfrA12

CCTCTACCGAACCGTCACACA

GCGACAGCGTTGAAACAACTAC

84

sul1

CAGCGCTATGCGCTCAAG

ATCCCGCTGCGCTGAGT

128

sul2

TCCGATGGAGGCCGGTATCTGG

CGGGAATGCCATCTGCCTTGAG

101

sul3

GCCGATGAGATCAGACGTATTG

CGCATAGCGCTGGGTTTC

189

strA

AATGAGTTTTGGAGTGTCTCAACGTA

AATCAAAACCCCTATTAAAGCCAAT

147

strB

GCTCGGTCGTGAGAACAATCT

CAATTTCGGTCGCCTGGTAGT

100

mcr-1

CGGTCAGTCCGTTTGTTC

CTTGGTCGGTCTGTAGGG

308

mcr-2

TGTTGCTTGTGCCGATTGGA

AGATGGTATTGTTGGTTGCTG

562

mcr-3

TTGGCACTGTATTTTGCATTT

TTAACGAAATTGGCTGGAACA

542

mcr-4

ATTGGGATAGTCGCCTTTTT

TTACAGCCAGAATCATTATCA

487

mcr-5

ATGCGGTTGTCTGCATTTATC

TCATTGTGGTTGTCCTTTTCTG

1644

tetA-01

GCTGTTTGTTCTGCCGGAAA

GGTTAAGTTCCTTGAACGCAAACT

62

tetA-02

CTCACCAGCCTGACCTCGAT

CACGTTGTTATAGAAGCCGCATAG

100

tetB-01

AGTGCGCTTTGGATGCTGTA

AGCCCCAGTAGCTCCTGTGA

62

tetB-02

GCCCAGTGCTGTTGTTGTCAT

TGAAAGCAAACGGCCTAAATACA

100

tetC-01

CATATCGCAATACATGCGAAAAA

AAAGCCGCGGTAAATAGCAA

77

tetC-02

ACTGGTAAGGTAAACGCCATTGTC

ATGCATAAACCAGCCATTGAGTAAG

104

tetD-01

TGCCGCGTTTGATTACACA

CACCAGTGATCCCGGAGATAA

85

tetD-02

TGTCATCGCGCTGGTGATT

CATCCGCTTCCGGGAGAT

100

tetE

TTGGCGCTGTATGCAATGAT

CGACGACCTATGCGATCTGA

73

tetG-01

TCAACCATTGCCGATTCGA

TGGCCCGGCAATCATG

92

tetG-02

CATCAGCGCCGGTCTTATG

CCCCATGTAGCCGAACCA

139

tetM-01

CATCATAGACACGCCAGGACATAT

CGCCATCTTTTGCAGAAATCA

100

tetM-02

TAATATTGGAGTTTTAGCTCATGTTGATG

CCTCTCTGACGTTCTAAAAGCGTATTAT

146

tetM-03

GCAATTCTACTGATTTCTGC

CTGTTTGATTACAATTTCCGC

185

floR-01

ATTGTCTTCACGGTGTCCGTTA

CCGCGATGTCGTCGAACT

60

catA1

GGGTGAGTTTCACCAGTTTTGATT

CACCTTGTCGCCTTGCGTATA

100

blaACC

CACACAGCTGATGGCTTATCTAAAA

AATAAACGCGATGGGTTCCA

67

blaCMY

CCGCGGCGAAATTAAGC

GCCACTGTTTGCCTGTCAGTT

107

blaCTX-M-01

GGAGGCGTGACGGCTTTT

TTCAGTGCGATCCAGACGAA

91

blaDHA

TGGCCGCAGCAGAAAGA

CCGTTTTATGCACCCAGGAA

120

blaIMP-01

AACACGGTTTGGTGGTTCTTGTA

GCGCTCCACAAACCAATTG

100

blaIMP-02

AAGGCAGCATTTCCTCTCATTTT

GGATAGATCGAGAATTAAGCCACTCT

232

blaIMP-03

GGAATAGAGTGGCTTAATTC

GGTTTAACAAAACAACCACC

71

blaKPC-02

CAGCTCATTCAAGGGCTTTC

GGCGGCGTTATCACTGTATT

195

blaKPC-03

GCCGCCGTGCAATACAGT

GCCGCCCAACTCCTTCA

59

blaSHV-01

TCCCATGATGAGCACCTTTAAA

TTCGTCACCGGCATCCA

69

mexF

CCGCGAGAAGGCCAAGA

TTGAGTTCGGCGGTGATGA

287

16S-01

GGGTTGCGCTCGTTGC

ATGGYTGTCGTCAGCTCGTG

60

16S-02

CCTACGGGAGGCAGCAG

ATTACCGCGGCTGCTGGC

195

rpoB

CGAACATCGGTCTGATCAACTC

GTTGCATGTTCGCACCCAT

359

  1. The quality of Figure 1, 2 and 3 is very poor. Please provide high quality of Figures. Response: the figures are modified.

  1. What are the three distinct populations for quinolones which the current study claims to find? From section 4. Discussion MIC distributions and the MIC50 and MIC90 values calculated showed a few differences compared to a previous study [31], with our study finding three distinct populations for quinolones and much higher values for quinolones and OXY.

Response: MIC distributions and the MIC50 and MIC90 values calculated showed a few differences compared to a previous study [31], with our study finding three distinct populations for quinolones (FLUQ: <0.125; 0.25- 2; > 4 µg/ml/ OA: <0.032; 0.064- 1; > 2 µg/ml/ ENRO: <0.032; 0.064- 0.25; > 0.5 µg/ml) and much higher values for quinolones and OXY.

  1. Language of text may be simplified to provide better clarity for the readers. As an example, in section 2.6 Biofilms samples “One biofilm surface was taken for the bacteriological analysis of cumulative previous months, and another biofilm surface was also collected from a previous month and then replaced with the biofilm surface of the following month. Each biofilm plate was placed in a sterile

Response: the sentence is modified.

 One biofilm surface was taken for the bacteriological analysis of cumulative previous months. Another biofilm surface was also collected from a previous month and then replaced with the biofilm surface of the following month.  Each biofilm plate was placed in a sterile bottle filled with the corresponding pond water bottle filled with the corresponding pond water.

  1. Few grammatical errors were noticed. As an example In lines from section 4. Discussion “Although 105 out 211 Aeromonas strains (49%) were consider as NWT for the three quinolones (FLUQ, OA and ENRO)”.

Response: the sentence is corrected.

Although 105 out 211 Aeromonas strains (49%) were considered as NWT for the three quinolones (FLUQ, OA and ENRO).

Reviewer 2 Report

Antibiotic resistance emergence is an actual and on-trend topic in both human and veterinary medicine. And in this sense, the manuscript gives interesting and valuable information on the establishing and development of antibiotic resistance in rainbow trout aquaculture, and draws opportune and helpful conclusions that, no doubt, will be of great help for the use of antibiotics in the very near future. However, there could be some confusion as in the tittle, and later in the text, authors only refer to Aeromonas in general, although they only tested strains of Aeromonas salmonicida subsp. salmonicida.  Therefore, it might be advisable that they change references to the genus in general, Aeromonas, to the real aeromonad representative they have tested (A. salmonicida subsp. salmonicida).

On the other hand, there are also a couple of other things that should be corrected, although these are much less important, and do not jeopardise the correct understanding of the text:

  1. In results, at the end of the second paragraph of “3.2 Antimicrobial susceptibility”, authors say “…presented the most difference…”, and maybe it would be more correct “…presented the biggest (or highest) difference…”;
  2. In discussion: all “et al.” should be in italics.

Author Response

Dear reviewer,

Thank you for your letter and the opportunity to revise our paper. The suggestions offered have been immensely helpful, and we also appreciate your insightful comments on revising the manuscript and other aspects of the paper.

I have included your comments immediately after this letter and responded to them individually indicating exactly how we addressed each concern or problem and describing the changes we have made. The changes are inserted and also marked in “Track Changes” in the revised paper. The revised manuscript is also submitted online by using the Editorial Manager system.

Comments:

  • There could be some confusion as in the tittle, and later in the text, authors only refer to Aeromonas in general, although they only tested strains of Aeromonas salmonicida salmonicida. Therefore, it might be advisable that they change references to the genus in general, Aeromonas, to the real aeromonad representative they have tested (A. salmonicida subsp. salmonicida).

Response: This study did not characterize Aeromonas strains. Therefore, we have tested Aeromonas in general including A. salmonicida subsp. salmonicida. We are only sure of the presence of A. salmonicida subsp. salmonicida on fish farm regarding to clinical and laboratory analysis.

Title is modified to clarify the subject of study:

Antimicrobial susceptibility profiles and resistance genes in genus Aeromonas spp. isolated from environment and rainbow trout of two fish farms in France

In manuscript text: Material and methods:

Aeromonas spp. isolation and identification

  1. In results, at the end of the second paragraph of “3.2 Antimicrobial susceptibility”, authors say “…presented the most difference…”, and maybe it would be more correct “…presented the biggest (or highest) difference…”;

Response: Oxolinic acid (OA) and flumequine (FLUQ) showed five and six dilutions, respectively, and trimethoprim-sulfamethoxazole (TMP) presented the highest difference with seven dilutions between MIC50 and MIC90 values. 

  1. In discussion: all “et al.” should be in italics.

Response: corrected.
